# Impact of Baseline Versus Intercurrent Steroids Administration on Upfront Chemo-Immunotherapy for Advanced Non-Small Cell Lung Cancer (NSCLC)

**DOI:** 10.3390/ijms231810292

**Published:** 2022-09-07

**Authors:** Andrea De Giglio, Marta Aprile, Alessandro Di Federico, Francesca Sperandi, Barbara Melotti, Francesco Gelsomino, Andrea Ardizzoni

**Affiliations:** 1Department of Experimental, Diagnostic and Specialty Medicine, University of Bologna, 40138 Bologna, Italy; 2Medical Oncology, IRCCS Azienda Ospedaliero-Universitaria di Bologna, 40138 Bologna, Italy

**Keywords:** non-small cell lung cancer, immunotherapy, chemo-immunotherapy, steroids, concomitant medications

## Abstract

The impact of baseline versus intercurrent steroids on the efficacy of upfront chemotherapy plus pembrolizumab (CT-ICI) for advanced non-small cell lung cancer (NSCLC) patients is unclear. We conducted a retrospective study on metastatic NSCLC patients treated with upfront CT-ICI at our institution between March 2020 and December 2021. The use of steroids was considered as the administration of at least 10 mg of prednisone equivalent. Of 101 patients, 36 (35.6%) received steroid therapy at baseline, and 18 (17.8%) started steroids on treatment. Overall, median progression-free survival (mPFS) was 6.5 months (95% CI, 5.9–8.9) and median overall survival (mOS) was 18.2 months (95% CI, 8.9-NR). Patients taking baseline steroids had significantly shorter survival than those not taking them and those assuming intercurrent steroids (mPFS 5.0 vs. 9.2 vs. 7.3 months, *p* < 0.001; mOS 7.0 months vs. not reached, *p* < 0.001). Baseline steroids were significantly associated with poorer survival outcomes in the multivariate model (OS HR 2.94, *p* = 0.02; PFS HR 3.84, *p* > 0.001). Conversely, intercurrent prescription did not reach a significant value regardless of other pivotal variables included in the model. Baseline steroid administration was associated with a detrimental effect on survival outcomes in NSCLC patients treated with CT-ICI. The role of intercurrent steroid administration should be further explored in larger studies.

## 1. Introduction

Over the last decade, immune-checkpoint inhibitors (ICI) gained a central role in the upfront treatment of metastatic non-oncogene addicted non-small-cell lung cancer (NSCLC). Antibodies targeting the programmed death-1 (PD-1) receptor, alone or combined with other molecules, improved survival outcomes compared to platinum-based chemotherapy regimens [1,2,3,4]. Programmed death ligand-1 (PD-L1) expression on tumor cells is currently the only biomarker used to guide treatment strategy. Since the results of the phase 3 trials Keynote 024 and Keynote 042, pembrolizumab monotherapy was approved as first-line therapy for PD-L1 1% NSCLC by the US Food and Drug Administration and for PD-L1 50% population by the European Medicines Agency [1,2]. More recently, phase 3 Keynote 189 and Keynote 407 trials, in non-squamous and squamous advanced NSCLC, respectively, reported significantly improved overall survival (OS) and progression-free survival (PFS) with manageable toxicity when adding pembrolizumab to first-line histology driven chemotherapy, irrespective of PD-L1 expression [3,4]. As a result, chemo-immunotherapy (CT-ICI) combinations have become a standard of care as first-line therapy among patients with metastatic NSCLC without druggable molecular alterations, regardless of PD-L1 tumor expression. Selecting the best upfront treatment for PD-L1 positive advanced NSCLC still represents an open issue [5,6]. Different predictive factors of response to ICI have been studied, such as the neutrophil to lymphocyte ratio, age, performance status, and concomitant therapies [7,8,9,10,11]. Steroids play a pivotal role in managing patients affected by NSCLC due to frequently associated cancer-related conditions such as dyspnea, pain, brain metastases, spinal cord compression, treatment-related adverse events. Baseline use of 10 mg of prednisone equivalents daily demonstrated a detrimental impact on objective response rate (ORR), OS, and PFS in retrospective studies among advanced NSCLC patients treated with single-agent immunotherapy in both first and second/third-line therapy [12,13,14,15]. However, current evidence suggests that corticosteroids reduce the survival benefits of immunotherapy only when administered for cancer-related symptoms, but their impact is not significant when cancer-unrelated conditions, such as immune-related adverse events (irAEs), bronchopneumonia (COPD) reactivation, or rheumatological diseases, represent the reason of prescription [16,17,18].

As systemic immunosuppressive therapy has been an exclusion criterion in clinical trials of CT-ICI combinations and no real-world data have been published in this field, we explored the impact of baseline and intercurrent corticosteroid administration (excluding those prescribed for chemotherapy pre- or post-medication) on survival outcomes of advanced NSCLC patients treated with first-line CT-ICI.

## 2. Results

A total of 101 consecutive patients received a first-line CT-ICI in the considered timeframe. The median age was 69 years (95% CI, 63.3–67.7). 62.4% of patients were male and 88.1% were affected by nonsquamous NSCLC. 89.1% were former or current smokers, and 89.1% had an Eastern Cooperative Oncology Group performance status (ECOG PS) of 0 or 1. Patients affected by nonsquamous NSCLC received 4 cycles of carboplatin (AUC5), pemetrexed (500 mg/m^2^) and pembrolizumab (200 mg flat dose) every 3 weeks, followed by maintenance therapy with pemetrexed (500 mg/m^2^) and pembrolizumab (200 mg flat dose) every 3 weeks. Patients affected by squamous NSCLC received 4 cycles of carboplatin (AUC6), paclitaxel (200 mg/m^2^), and pembrolizumab (200 mg, flat dose) every 3 weeks, followed by maintenance therapy with pembrolizumab (200 mg, flat dose) every 3 weeks.

Baseline characteristics are summarized in Table 1.

According to baseline steroid prescription, no relevant distribution imbalances were observed except for patients with brain metastasis at diagnosis who were more likely to take corticosteroids.

36 patients (35.6%) were receiving an oral or parenteral corticosteroid prescription of at least 10 mg of prednisone equivalent before starting CT-ICI treatment. The median daily dose was 27 mg (interquartile range, IQR, 20–50). Indications for steroid prescription were: symptomatic brain metastases (*n* = 20; 55.5%); dyspnea or other respiratory symptoms (*n* = 9; 25%); cancer-related pain (*n* = 5; 13.5%); fatigue (*n* = 2; 5%). On treatment steroids have been prescribed to 18 patients (17.8%) with a median daily intake of 25 mg (IQR 25–50). Reasons for steroid use among them were: treatment of therapy-related adverse effects (*n* = 13; 72.2%; 8 immunotherapy-related, 5 chemotherapy-related); cancer-related symptoms (*n* = 3; 16.7%), namely symptomatic brain metastases (*n* = 1) and dyspnea (*n* = 2); intercurrent diseases represented by pneumonia and sarcoidosis (*n* = 2; 11.1%).

Globally, the median OS was 18.2 months (95% CI, 8.9-not reached, NR). The median follow-up time was 8.3 months (IQR, 3.9–14.3). Patients receiving steroids at baseline experienced a median OS of 7.0 months (95% CI, 4.2-NR) in comparison to a not-reached median OS for patients not receiving steroids (95% CI, 8.35-NR) or those assuming intercurrent steroids (95% CI, NR-NR) (*p* < 0.001) (Figure 1). 

The baseline steroid prescription was associated with an increased risk of death both in the univariate analysis (HR 2.66, 95% CI, 1.29–5.48, *p* = 0.008) and in the multivariate analysis (HR 2.94, 95% CI, 1.18–7.31, *p* = 0.02), after adjusting for age, sex, histology, smoking status, ECOG PS, number of metastatic sites, brain and liver involvement, and intercurrent steroid prescription. Patients taking steroids intercurrently experienced a non- significantly reduced death risk. Brain metastasis at baseline was associated with reduced survival within the univariate but not multivariate model (HR 1.26, 95% CI, 0.53–3.03, *p* = 0.602) (Table 2).

The median PFS within the whole cohort was 6.5 months (95% CI, 5.9–8.9). The median PFS was 5.0 months (95% CI, 2.4–6.4) among patients with baseline steroid intake versus 9.2 months (95% CI, 6.1-NR) among those with intercurrent intake and 7.3 months (95% CI, 6.9-NR) among those never treated with steroids (*p* < 0.001) (Figure 2). 

Baseline steroid intake was associated with an increased risk of disease progression in both univariate (HR: 2.74; 95% CI, 1.46–4.79, *p* = 0.001) and multivariate analyses (HR 3.84, 95% CI, 1.82–8.08, *p* < 0.001). Squamous histology was an independent negative prognostic factor for disease progression confirmed in the multivariate assessment (HR 2.57, 95% CI, 1.14–5.79, *p* = 0.023). No other variables included in the model showed a significant independent association with progression risk (Table 2).

## 3. Discussion

To our knowledge, this is the first report analyzing the prognostic role of corticosteroid intake during first-line CT-ICI in patients with advanced NSCLC. This information, unfortunately, cannot be extrapolated from randomized CT-ICI trials, since these studies did not allow the enrollment of patients requiring steroid use and no formal analyses have been performed on patients assuming steroids during the treatment. The warning on the steroid introduction during an immune-stimulating treatment was based on the clinical evidence of increased infectious disease rate for patients taking at least 10 mg of prednisone equivalents daily [19]. Thus, patients overcoming this threshold have been excluded from clinical trials regardless of clinical indication. Herein, we provide retrospective evidence that baseline intake of steroids is associated with hampered survival outcomes while, on the contrary, intercurrent steroid intake does not have a survival impact independently from other relevant clinical variables. Several retrospective studies have explored the impact of steroids on single-agent ICI in lung cancer patients either in first or further treatment lines. The prognostic impact of baseline steroid intake on survival outcomes was investigated among 640 advanced NSCLC patients receiving ICI. Baseline steroids were associated with significantly reduced PFS (HR: 1.31; *p* < 0.03) and OS (HR: 1.66; *p* < 0.001) [12]. Patients with poor ECOG PS or active brain metastasis were more likely to take steroids. Another retrospective work confirmed the negative association between baseline steroid administration and survival among PD-L1-high NSCLC patients treated with first-line pembrolizumab, with a 2.3-fold increased risk of death [13]. Comparably, we evidenced a significant increase in the risk of disease progression (HR 3.84, 95% CI, 1.82–8.08, *p* < 0.001) and death (HR 2.94, 95% CI, 1.18–7.31, *p* = 0.02) after adjusting for pivotal variables. Two observational studies evidenced that an early introduction of steroids during immunotherapy was independently associated with poor outcomes. Drakaki et al. showed that steroids introduced within the first month of ICI treatment negatively impacted survival outcomes in a large cohort of 862 advanced NSCLC patients [14]. Fucà et al. speculated that the detrimental ICI results found among 35 patients receiving steroids in the first 28 days of treatment were related to the variation of circulating lymphocyte subpopulations [15]. Even if the stimulation of the glucocorticoid receptor is predominantly associated with a depressed innate and adaptive immune response, comprehensive genome analyses suggested that steroids produce a precocious but short-lasting immune activation and theorized a biphasic model of response under steroid treatment [20]. According to this model, in the absence of endogenous steroids, the immune response would be delayed but long-lasting [20]. Interestingly, analyzing the reasons for steroid prescription and distinguishing between cancer-related (brain edema, dyspnea, fatigue) and unrelated (irAEs, COPD reactivation, rheumatological disease) medical conditions, the introduction of steroids seemed not to influence treatment outcomes. Ricciuti et al. demonstrated that NSCLC patients on ICI treatment receiving baseline ≥10 mg prednisone equivalents for cancer-unrelated conditions and patients taking <10 mg prednisone equivalents for any reason had longer mPFS (4.6 vs. 3.4 vs. 1.4 months; *p* < 0.001) and mOS (10.7 vs. 11.2 vs. 2.2 months; *p* < 0.001) compared with patients taking ≥10 mg prednisone equivalents for cancer-related conditions [17]. Likewise, a monocentric experience explored the outcomes of 413 advanced NSCLC patients under ICI single-agent treatment, stratifying survival outcomes according to timing (baseline vs. intercurrent) and reason (palliative vs. non-palliative) for steroid prescription. Intercurrent steroid introduction for cancer-unrelated symptoms was not associated with poor survival [18]. Conversely, intercurrent administration for cancer palliation was independently associated with poor PFS and OS (*p* < 0.0001). Our analysis could not efficiently explore the impact of cancer-related and unrelated clinical indications due to the retrospective nature of the study and its limited sample size. Nevertheless, we should consider that all patients treated with CT-ICI received steroids premedication according to the chemotherapy regimen administered. Moreover, all baseline steroid-treated patients included in our report were prescribed only for cancer-related indications. De facto, our analysis demonstrated that the steroid intake for palliative reasons at baseline constituted an independent risk factor for disease progression and death despite the routine premedication prescription. Belonging to an unfavorable prognostic group may affect the reliability of the pure effect of steroids on ICI or CT-ICI efficacy. Mainly, the baseline steroid intake was more frequently associated with the presence of symptomatic brain metastases. Remarkably, our analysis did confirm the negative prognostic relevance of brain metastasis at diagnosis in the univariate but not in the multivariable model, probably due to the limited sample size. Concerning the intercurrent administration, the inclusion of a small subset of patients receiving steroids for both palliative and non-palliative indications did not allow us to draw a robust conclusion in this field of debate. Analogously, 8 out of 18 patients received intercurrent steroids for the treatment of immune-related adverse events. The positive prognostic value of immune-related adverse events under ICI [21] is well established but cannot be evaluated in this analysis due to the limited subgroup.

This setting should be further investigated in a larger population, considering the risk of immortal time bias. Overall, the absence of translational studies exploring the biological rationale of steroids and immune system interplay under ICIs and the studies dissecting the clinical indication for steroid prescription reduce the strength of these retrospective findings. Nevertheless, the novelty of the population investigated for this research question, and the number of key variables included in the multivariable assessment, represented the strengths of the present investigation.

## 4. Materials and Methods

We conducted a single-center, observational retrospective study including consecutive patients treated according to clinical practice with upfront CT-ICI for advanced non-oncogene-addicted NSCLC between March 2020 and December 2021 at the Sant’Orsola-Malpighi University Hospital (Bologna, Italy). Medical records have been investigated to extract clinical and biological data. The following variables have been collected: age, gender, comorbidities, concomitant therapies, tumor histology, molecular characterization, anticancer treatments, Eastern Cooperative Oncology Group (ECOG) performance status (PS) at baseline, steroid information (type, dosage, prescription data), radiological findings at baseline and during the follow-up, last follow-up, cause of death, date of death. We then explored the impact of steroid prescription on survival outcomes, considering the prescription of oral or parenteral 10 mg of prednisone equivalents, excluding prescriptions for chemotherapy pre or post-medication. We evaluated the impact of baseline and intercurrent steroid prescription. Patients on steroid treatment within 24 h of the first cycle of CT-ICI were included in the baseline intake group. Patients prescribed during the treatment but not taking baseline steroids were included in the intercurrent intake group. This study received approval from the local Ethics Committee (approval no. 2381/2019), and all patients signed written informed consent. The study was conducted in accordance with the Declaration of Helsinki (1964).

### Statistical Methods

Continuous and categorical variables were described as median values and proportions. T-test (or ANOVA if needed) and chi2-test (or Fisher’s exact test, if required) were performed to assess inference between means and proportions. The normality of the distribution of variables of interest was explored via Shapiro test. The primary endpoint was OS, defined as the time from treatment start to death from any cause. The secondary endpoint was PFS, defined as the time occurring from treatment start to the first radiological or clinical disease progression, or death from any cause. The data cut-off was December 2021; after that, patients still alive were censored at last contact. Survival times have been estimated through the Kaplan-Meier method, and survival outcomes were compared through the Log Rank Test. The reverse Kaplan-Meier method allowed us to calculate the median time of follow-up. The relationship between clinical or biological variables and survival outcomes was explored in a stepwise fashion through a univariate and then multivariate analysis using a Cox model regression for both survival endpoints. A *p*-value ≤ 0.05 was considered statistically significant. Statistical analyses were performed with the open-source software R-Studio, version 1.4.1717, using the following packages: ‘dplyr’, ‘prodlim’, ‘survminer’, ‘survMisc’,’finalfit’.

## 5. Conclusions

Indeed, more extensive prospective studies are required to assess the real impact of the steroids according to various clinical indications both at baseline and during the CT-ICI regimens. Further translational studies are warranted to investigate the biological rationale of the multiple interactions between ICIs, immune system, and immune-modulating agents. Nevertheless, our experience suggests that baseline steroid intake constitutes a poor risk factor for CT-ICI efficacy and, therefore, this information should be considered when choosing this treatment strategy in advanced NSCLC. On the contrary, intercurrent steroids, when needed, can be prescribed without the worry of hampering CT-ICI efficacy.

## Figures and Tables

**Figure 1 ijms-23-10292-f001:**
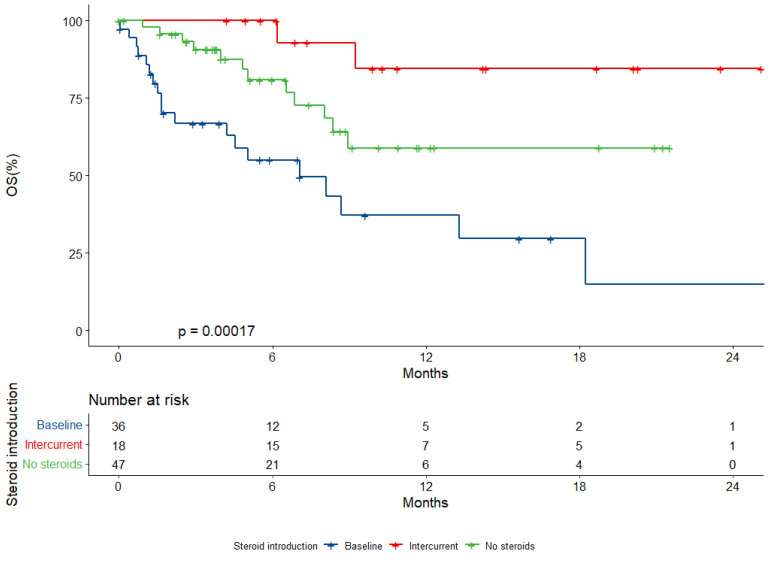
Overall survival (OS) according to baseline steroid prescription.

**Figure 2 ijms-23-10292-f002:**
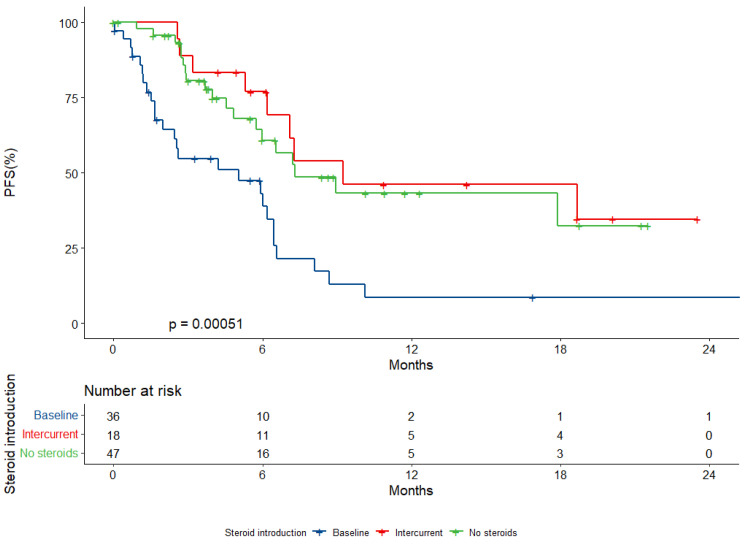
Progression-free survival (PFS) according to baseline steroid prescription.

**Table 1 ijms-23-10292-t001:** Baseline characteristics according to baseline steroid prescription.

		Baseline Steroids			
		No (%)	Yes (%)	Total (%)	*p*-Value
Age	≤65	25 (38.5)	15 (41.7)	40 (39.6)	0.918
	>65	40 (61.5)	21 (58.3)	61 (60.4)	
Sex	Female	24 (36.9)	14 (38.9)	38 (37.6)	1.000
	Male	41 (63.1)	22 (61.1)	63 (62.4)	
Histology	Nonsquamous	57 (87.7)	32 (88.9)	89 (88.1)	1.000
	Squamous	8 (12.3)	4 (11.1)	12 (11.9)	
Smoking status	former smoker	39 (60.0)	22 (61.1)	61 (60.4)	0.693
	never smoker	6 (9.2)	5 (13.9)	11 (10.9)	
	smoker	20 (30.8)	9 (25.0)	29 (28.7)	
ECOG PS	0–1	61 (93.8)	29 (80.6)	90 (89.1)	0.085
	2	4 (6.2)	7 (19.4)	11 (10.9)	
Antibiotic prescription	No	31 (57.4)	12 (38.7)	43 (50.6)	0.151
	Yes	23 (42.6)	19 (61.3)	42 (49.4)	
n. of metastatic sites	1 to 3	49 (77.8)	24 (66.7)	73 (73.7)	0.331
	at least 4	14 (22.2)	12 (33.3)	26 (26.3)	
Bone met.	No	36 (57.1)	23 (63.9)	59 (59.6)	0.656
	Yes	27 (42.9)	13 (36.1)	40 (40.4)	
Brain met.	No	54 (85.7)	13 (36.1)	67 (67.7)	<0.001
	Yes	9 (14.3)	23 (63.9)	32 (32.3)	
Liver met.	No	54 (85.7)	30 (83.3)	84 (84.8)	0.979
	Yes	9 (14.3)	6 (16.7)	15 (15.2)	

Abbreviations: n., number; met., metastasis; ECOG PS, Eastern Cooperative Oncology Group performance status.

**Table 2 ijms-23-10292-t002:** Univariate and multivariate analysis for progression-free survival (PFS) and overall survival (OS).

		All (%)	PFS		OS	
			HR (Univariable)	HR (Multivariable)	HR (Univariable)	HR (Multivariable)
Age	≤65	40 (39.6)	-	-	-	-
	>65	61 (60.4)	1.19 (0.69–2.08, *p* = 0.531)	1.20 (0.64–2.23, *p* = 0.570)	1.13 (0.56–2.27, *p* = 0.737)	0.82 (0.37–1.83, *p* = 0.629)
Sex	Female	38 (37.6)	-	-	-	-
	Male	63 (62.4)	1.00 (0.58–1.73, *p* = 0.990)	1.04 (0.55–1.97, *p* = 0.900)	2.00 (0.93–4.30, *p* = 0.078)	2.37 (0.93–6.02, *p* = 0.071)
Histology	Nonsquamous	89 (88.1)	-	-	-	-
	Squamous	12 (11.9)	2.30 (1.11–4.77, *p* = 0.026)	2.57 (1.14–5.79, *p* = 0.023)	1.97 (0.81–4.81, *p* = 0.136)	2.57 (0.98–6.73, *p* = 0.055)
Smoking status	Former smoker	61 (60.4)	-	-	-	-
	Never smoker	11 (10.9)	0.55 (0.21–1.42, *p* = 0.217)	0.33 (0.10–1.03, *p* = 0.056)	0.34 (0.08–1.48, *p* = 0.151)	0.33 (0.06–1.89, *p* = 0.212)
	Smoker	29 (28.7)	0.91 (0.51–1.64, *p* = 0.760)	0.85 (0.45–1.60, *p* = 0.606)	0.70 (0.32–1.53, *p* = 0.371)	0.79 (0.34–1.84, *p* = 0.583)
ECOG PS	0–1	90 (89.1)	-	-	-	-
	2	11 (10.9)	1.59 (0.75–3.37, *p* = 0.228)	1.60 (0.59–4.30, *p* = 0.356)	1.80 (0.69–4.67, *p* = 0.228)	1.75 (0.51–6.00, *p* = 0.373)
n. of metastatic sites	≤3	73 (73.7)	-	-	-	-
	>4	26 (26.3)	1.18 (0.63–2.20, *p* = 0.612)	1.15 (0.53–2.50, *p* = 0.733)	1.78 (0.84–3.77, *p* = 0.129)	1.69 (0.69–4.17, *p* = 0.254)
Brain met.	No	67 (67.7)	-	-	-	-
	Yes	32 (32.3)	1.32 (0.76–2.30, *p* = 0.327)	0.73 (0.36–1.49, *p* = 0.390)	2.24 (1.13–4.44, *p* = 0.021)	1.26 (0.53–3.03, *p* = 0.602)
Liver met.	No	84 (84.8)	-	-	-	-
	Yes	15 (15.2)	2.30 (1.17–4.53, *p* = 0.016)	1.99 (0.88–4.51, *p* = 0.099)	1.23 (0.47–3.20, *p* = 0.673)	1.34 (0.43–4.20, *p* = 0.614)
Steroid introduction	No steroids	47 (46.5)	-	-	-	-
	Baseline	36 (35.6)	2.64 (1.46–4.79, *p* = 0.001)	3.84 (1.82–8.08, *p* < 0.001)	2.66 (1.29–5.48, *p* = 0.008)	2.94 (1.18–7.31, *p* = 0.020)
	Intercurrent	18 (17.8)	0.85 (0.38–1.88, *p* = 0.685)	1.25 (0.53–2.96, *p* = 0.607)	0.27 (0.06–1.22, *p* = 0.088)	0.35 (0.07–1.67, *p* = 0.188)

Abbreviations: HR, hazard ratio; ECOG PS, Eastern Cooperative Oncology Group performance status; n., number.; met., metastasis.

## Data Availability

The datasets generated during and/or analyzed during the current study are available from the corresponding author on reasonable request.

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
