# Peer review of "Impact of Baseline Versus Intercurrent Steroids Administration on Upfront Chemo-Immunotherapy for Advanced Non-Small Cell Lung Cancer (NSCLC)"

_ijms, 2022, doi:10.3390/ijms231810292_

Round 1
Reviewer 1 Report
Steroid use have been associated with poorer outcome for patients who received IO's. The current study looks at patients who had simultaneous chemo-IO. The major weakness of the study is the small size and thus limited power. I also think the authors need to address or at least discuss more in depth the indication bias for steroids, IE that i is probably the sickest lung cancer patients that are on steroids at baseline and PS may not adequately adjust for this. I therefore am not sure that one can conclude as the author do that steroids at baseline is associated with a worse outcome. I agree with the authors that this complex interaction may not be difficult to study in a retrospective study so it would be interesting if the authors discuss alternative designs to study the association. I also agree with the authors that steroids administered after the start of treatment is a different thing but it is noteworthy that these patients actually have a better out come.
Author Response
Response to reviewer 1 comments
We are grateful to the reviewer for the comments received.
Reviewer #1:
Point 1 : Steroid use have been associated with poorer outcome for patients who received IO's. The current study looks at patients who had simultaneous chemo-IO. The major weakness of the study is the small size and thus limited power.
Point 1 response: To improve the research's statistical methodology, we analyzed the prognostic impact of steroid introduction according to three categories: baseline steroids, intercurrent steroids, and no steroids. We performed new Log-rank tests and cox regression analyses confirming the prognostic negative role of baseline assumption for death and progression risks. We consider this paper a proof of concept to pave the way for more extensive studies.
Point 2: I also think the authors need to address or at least discuss more in depth the indication bias for steroids, IE that i is probably the sickest lung cancer patients that are on steroids at baseline and PS may not adequately adjust for this. I therefore am not sure that one can conclude as the author do that steroids at baseline is associated with a worse outcome. I agree with the authors that this complex interaction may not be difficult to study in a retrospective study so it would be interesting if the authors discuss alternative designs to study the association. I also agree with the authors that steroids administered after the start of treatment is a different thing but it is noteworthy that these patients actually have a better out come.
Response to point 2: In the discussion section, we added the following statement: ‘Overall, the absence of translational studies exploring the biological rationale of steroids and immune-system interplay under ICIs and the studies dissecting the clinical indication for steroids prescription reduce the strength of these retrospective findings'.
In the conclusion section we declared : ‘Indeed, more extensive prospective studies are required to assess the real impact of the steroids according to various clinical indications both at baseline and during the CT-ICI regimens. Further translational studies are warranted to investigate the biological rationale of the multiple interactions between ICIs, immune-system, and immune-modulating agents. Nevertheless, our experience suggests that baseline steroid intake constitutes a poor risk factor for CT-ICI efficacy and, therefore, this information should be considered when choosing this treatment strategy in advanced NSCLC. On the contrary, intercurrent steroids, when needed, can be prescribed without the worry of hampering CT-ICI efficacy.’
Reviewer 2 Report
In the manuscript, the Authors evaluated the impact of baseline and intercurrent steroid administration on treatment outcomes in NSCLC patients treated with chemo-immunotherapy. They found that the baseline steroids resulted in shorter PFS and OS while intercurrent steroid administration prolonged OS in the studied cohort. The topic is interesting; however, I have some reservations about the presented conclusions, which might be misleading. Please, find the detailed comments below.
Major issues:
· I am not convinced about the conclusion that patients taking steroids throughout the treatment presented longer OS. I wonder why the survival of these patients was compared to the entire study population, not to the “control group”, i.e., patients not taking steroids at all. If baseline steroids indeed shortened OS, a group comprising “baseline steroids” + “control group” had shorter OS than the “control group”. This could explain the observed difference. In my opinion, to show the steroid effect on the treatment outcomes, both steroid groups should be compared to the “control group”. I can see a rationale standing behind separate comparisons for baseline/intercurrent steroids; however, with this small cohort (36 baseline steroids, 18 on-treatment steroids, and 47 controls), “neutral patients” (patients w/o steroid treatment) are (unfortunately) only a small portion of the study population and “steroid patients” could have affected the results significantly. The Authors should create 3 groups for statistical purposes: “baseline steroids”, “intercurrent steroids”, and “no steroids” and this variable should be included in Kaplan-Meier and Cox regression analysis.
· Moreover, patients dosed with steroids during CT-ICI treatment usually received these steroids due to adverse effects (72%, line 93). Were they immune-related adverse events? If so, better treatment outcomes in those patients could result from more pronounced activation of the immune system, not from the on-treatment steroid intake. In such a case, the conclusion about steroid intake & better treatment outcomes (prolonged OS) would be completely wrong because it suggests that better treatment outcomes resulted from steroid intake (= it encourages administering steroids during the treatment). Have the Authors tried to include this variable (immune-related adverse effects) into the Cox regression model to exclude the confounding influence?
· Discussion should be divided into smaller sections because it is hard to follow.
· References do not seem to be cited correctly – please revise the entire manuscript; e.g., in line 159, the cited reference (15) does not refer to Drakaki et al.; in line 161, the cited reference (16) does not refer to Fuca et al.
Minor issues:
· Some editorial errors need to be corrected, e.g., mg/m2 – please place “2” in superscript.
· 95% CI is sometimes presented as … - NR and sometimes as … - NA.
· Table 1: I would suggest placing the “baseline steroids” header above “yes/ no/ total” for better clarity.
· IQR abbreviation should be introduced after the first use.
· Figure 3: There is no need to present a p-value with such a high accuracy – p<0.001 would be enough (this comment applies to all parts of the manuscript as well as other p-values).
· Table 2: I would suggest removing “doubled variables” from the table for better clarity; e.g., there is no need to present data for ≤65 and >65; moreover, the “All (%)” variable is repeated from Table 1, so I would suggest removing these two columns with “All (%)” variable from Table 2.
· Line 150: Please revise the presented data for possible typos because 1.7 vs. 1.8 does not seem to be statistically different.
· Lines 169 – 169: The sentence should be revised for better clarity.
· Line 227: Please double-check if Pearson correlation was indeed used in the statistics.
Author Response
Response to reviewer 2 comments
We thank the reviewer for the great contribution to our work.
Reviewer #2:
Point 1 I am not convinced about the conclusion that patients taking steroids throughout the treatment presented longer OS. I wonder why the survival of these patients was compared to the entire study population, not to the “control group”, i.e., patients not taking steroids at all. If baseline steroids indeed shortened OS, a group comprising “baseline steroids” + “control group” had shorter OS than the “control group”. This could explain the observed difference. In my opinion, to show the steroid effect on the treatment outcomes, both steroid groups should be compared to the “control group”. I can see a rationale standing behind separate comparisons for baseline/intercurrent steroids; however, with this small cohort (36 baseline steroids, 18 on-treatment steroids, and 47 controls), “neutral patients” (patients w/o steroid treatment) are (unfortunately) only a small portion of the study population and “steroid patients” could have affected the results significantly. The Authors should create 3 groups for statistical purposes: “baseline steroids”, “intercurrent steroids”, and “no steroids” and this variable should be included in Kaplan-Meier and Cox regression analysis.
Point 1 response:
We modified the project design following the reviewer's indications. We created 3 groups according to steroid introduction and re-performed the Log-Rank test and Cox regression analysis.
Point 2: Moreover, patients dosed with steroids during CT-ICI treatment usually received these steroids due to adverse effects (72%, line 93). Were they immune-related adverse events? If so, better treatment outcomes in those patients could result from more pronounced activation of the immune system, not from the on-treatment steroid intake. In such a case, the conclusion about steroid intake & better treatment outcomes (prolonged OS) would be completely wrong because it suggests that better treatment outcomes resulted from steroid intake (= it encourages administering steroids during the treatment). Have the Authors tried to include this variable (immune-related adverse effects) into the Cox regression model to exclude the confounding influence?
Response to point 2: The small sample size is the main weak point of the present research paper. As discussed, we could not perform appropriate analysis of cancer-related and unrelated indications (i.e., immune-related adverse events). Furthermore, the role of intercurrent introduction should be further investigated in a larger population considering the immortal time bias (that requires a landmark analysis to be reduced). In particular, the prognostic value of immune-related adverse events cannot be adequately explored due to the small subgroup (8 pts received intercurrent steroid for irAEs, 5 for chemotherapy-related adverse events).
To emphasize this passage, we added this point of weakness within the discussion: ‘Analogously, 8 out of 18 patients received intercurrent steroids for the treatment of immune-related adverse events. The positive prognostic value of immune-related adverse events under ICI is well established but cannot be evaluated due to the limited subgroup.’
Point 3: Discussion should be divided into smaller sections because it is hard to follow.
Response to point 3: We added a conclusion section to be clearer, following the IJMS guidelines.
Point 4: References do not seem to be cited correctly – please revise the entire manuscript; e.g., in line 159, the cited reference (15) does not refer to Drakaki et al.; in line 161, the cited reference (16) does not refer to Fuca et al.
Response to point 4: We revised and corrected all the cited references.
Minor issues:
- Some editorial errors need to be corrected, e.g., mg/m2 – please place “2” in superscript.
- 95% CI is sometimes presented as … - NR and sometimes as … - NA.
- Table 1: I would suggest placing the “baseline steroids” header above “yes/ no/ total” for better clarity.
- IQR abbreviation should be introduced after the first use.
- Figure 3: There is no need to present a p-value with such a high accuracy – p<0.001 would be enough (this comment applies to all parts of the manuscript as well as other p-values).
- Table 2: I would suggest removing “doubled variables” from the table for better clarity; e.g., there is no need to present data for ≤65 and >65; moreover, the “All (%)” variable is repeated from Table 1, so I would suggest removing these two columns with “All (%)” variable from Table 2.
- Line 150: Please revise the presented data for possible typos because 1.7 vs. 1.8 does not seem to be statistically different.
- Lines 169 – 169: The sentence should be revised for better clarity.
- Line 227: Please double-check if Pearson correlation was indeed used in the statistics.
Minor issues response: All issues have been fixed following the indications.
Round 2
Reviewer 1 Report
thank you for addressing the issues that were raised.